# Pharmacological Inhibition of the Asparaginyl Endopeptidase (AEP) in an Alzheimer’s Disease Model Improves the Survival and Efficacy of Transplanted Neural Stem Cells

**DOI:** 10.3390/ijms24097739

**Published:** 2023-04-23

**Authors:** Qing Cheng, Xiaoli Ma, Jingjing Liu, Xuemei Feng, Yan Liu, Yanxia Wang, Wenwen Ni, Mingke Song

**Affiliations:** Department of Pharmacology and Chemical Biology, Institute of Medical Sciences, Shanghai Jiao Tong University School of Medicine, 280 South Chongqing Road, Shanghai 200025, China

**Keywords:** Alzheimer’s disease, stem cell therapy, neural stem cells, amyloid-beta (Aβ), neuroinflammation, brain microenvironment, asparaginyl endopeptidase (AEP), cell transplantation, APP/PS1 mice, legumain

## Abstract

Stem-cell-based therapy is very promising for Alzheimer’s disease (AD), yet has not become a reality. A critical challenge is the transplantation microenvironment, which impacts the therapeutic effect of stem cells. In AD brains, amyloid-beta (Aβ) peptides and inflammatory cytokines continuously poison the tissue microenvironment, leading to low survival of grafted cells and restricted efficacy. It is necessary to create a growth-supporting microenvironment for transplanted cells. Recent advances in AD studies suggest that the asparaginyl endopeptidase (AEP) is a potential intervention target for modifying pathological changes. We here chose APP/PS1 mice as an AD model and employed pharmacological inhibition of the AEP for one month to improve the brain microenvironment. Thereafter, we transplanted neural stem cells (NSCs) into the hippocampus and maintained therapy for one more month. We found that inhibition of AEPs resulted in a significant decrease of Aβ, TNF-α, IL-6 and IL-1β in their brains. In AD mice receiving NSC transplantation alone, the survival of NSCs was at a low level, while in combination with AEP inhibition pre-treatment the survival rate of engrafted cells was doubled. Within the 2-month treatment period, implantation of NSCs plus pre-inhibition of the AEP significantly enhanced neural plasticity of the hippocampus and rescued cognitive impairment. Neither NSC transplantation alone nor AEP inhibition alone achieved significant efficacy. In conclusion, pharmacological inhibition of the AEP ameliorated brain microenvironment of AD mice, and thus improved the survival and therapeutic efficacy of transplanted stem cells.

## 1. Introduction

Alzheimer’s disease (AD) is a neurodegenerative disease, asymptomatic in the preclinical stage, with mild cognitive impairment in the prodromal stage and dementia particularly in the late stage [1]. The amyloid cascade hypothesis, which posits the deposition of amyloid-beta (Aβ) peptide in the brain as a core pathology, is the main basis for the biological understanding of AD, drug development and biomarker-based diagnosis [2,3]. From 1993 to 2014, only cholinesterase inhibitors and memantine were approved for the treatment of AD; however, these symptomatic therapeutics cannot cure AD dementia [4]. More recently, the new agents Oligomannate (GV-971), aducanumab and lecanemab were developed for therapeutic trials of AD patients, yet their exact therapeutic effects still need to be evaluated through accumulating more clinical evidence. To date, there are no definitively effective medications for AD [5,6,7,8]. 

Stem cells hold promising applications in regenerative medicine, therefore bringing new hope for modifying AD and other neurodegenerative diseases [9,10,11]. Preclinical and clinical studies of cell-based therapy have transplanted various types of stem cells into AD brains and obtained varying efficacy and even contradictory outcomes [12,13,14]. These engraftments include neural stem cells (NSCs) isolated from the brain, NSCs derived from embryonic stem cells (ESCs) or induced pluripotent stem cells (iPSCs), as well as mesenchymal stem cells (MSCs) [12,14,15]. Transplantation of NSCs has become one of the main lines of AD treatment studies. However, there are two major challenges existing in this research area. One difficulty is the acquisition and optimization of NSCs to fulfill the standards of Good Manufacturing Practice (GMP) for transplantation [16,17,18]. The second highlighted concern is the pathological microenvironment in the recipient brain that may impair the viability of grafted cells and restrict the therapeutic effect of NSCs [16,17,19,20,21]. 

The basal microenvironment in AD brains consists of neurotoxicity of the amyloid-beta (Aβ) oligomers and microglial neuroinflammation [22,23,24]. Persistent release of Aβ and inflammatory cytokines in the host brain is hazardous to implanted cells and leads to poor cell survival after transplantation. Sugaya et al. found that the differentiation capability of human NSCs was greatly suppressed after they were transplanted into brains of amyloid precursor protein (APP) transgenic mice [25,26]. In vitro, Aβ was reported to suppress proliferation and differentiation of cultured NSCs derived from human and rodent cerebral cortex, and even induce cellular apoptosis in these cells [27,28,29]. In AD mice brains, Aβ not only impaired adult hippocampal neurogenesis but also inhibited the migration and proliferation of NSCs in the subventricular zone (SVZ) [30,31]. The action of inflammatory milieu on the fate of NSCs is more complex than Aβ; both negative and positive actions of inflammation on NSCs have been documented. The proinflammatory cytokines TNF-α and IL-1β, and microglial activation, were reported to inhibit the survival and differentiation of NSCs, while SDF-1, IL-1α and IL-10 were reported to be beneficial for NSC migration and differentiation [19,32,33]. Fractalkine, also named C-X3-C Motif Chemokine Ligand 1 (CX3CL1), is produced by neurons, while its receptor CX3CR1 is mainly expressed on microglia. CX3CL1-CX3CR1 signaling was initially found to exhibit an anti-inflammatory effect in some AD pathology studies, but in other cases it instead incurred neurodegeneration [34,35]. To promote stem cell therapy of AD to thrive, it is of great importance to modulate Aβ and inflammation-associated milieu in host brains and construct a growth-supporting environment for NSC survival. 

Conceivably, suppressing Aβ and neuroinflammation prior to transplantation could improve the recipient brain microenvironment and support the survival of implanted NSCs. Recent accumulating evidence indicates that the lysosomal asparaginyl endopeptidase (AEP) or legumain can serve as an intervention target for reducing Aβ generation and inhibiting inflammatory responses [36,37,38]. The expression level and enzymatic activity of AEPs in AD and aging brains are progressively upregulated [39,40,41,42,43]. The activated AEP acts as a δ-secretase to cleave APP into amyloidogenic fragments, facilitating β-secretase to digest these APP fragments, thereupon feeding more peptides to γ-secretase to produce more Aβ [43]. In AD animal models, deletion or pharmacological inhibition of the AEP was found to reduce Aβ accumulation [39,40,43,44]. Meanwhile, animal model studies have reported AEP as an important player in brain neuroinflammation. Wu et al. found that traumatic brain injury (TBI) led to microglia activation and continuous release of inflammatory factors (IL-1β, IL-6, TNF-α) in mice, together with an increase in brain AEP activity [45]. Knockout of AEP gene significantly inhibited these TBI-induced inflammatory responses. In a mouse model of AD induced by intracerebroventricular injection of Aβ_1–42_, deletion of legumain gene (encoding AEP) decreased upregulation of IL-1β, IL-6 and TNF-α in the hippocampus [46]. 

Based on the above background introduction and translational questions, we hypothesized that pharmacological inhibition of the AEP could ameliorate tissue microenvironment of the recipients, which in turn could support survival of implanted NSCs. The present study employed a brain-penetrant AEP inhibitor to repress inflammatory response and Aβ generation in the brain of an AD model: APP/PS1 transgenic mice. Thereafter, NSCs derived from embryonic brains of mice were transplanted into the hippocampus of APP/PS1 mice that had received AEP inhibitor pre-treatment. We examined if this combination therapy strategy could improve the efficacy of NSC transplantation in this AD model.

## 2. Results

### 2.1. Mouse NSCs and Transfection with Green Fluorescent Protein (GFP)

We isolated NSCs from the developing brain of mouse embryos and cultured them in suspension to form neurospheres. When needed, we changed the suspension culture mode to the adherent monolayer culture to obtain dispersed NSCs (Figure 1a). Immunofluorescent staining of neurospheres revealed the presence of the neural stem cell markers Nestin and SOX2 (Appendix A). Additionally, we transfected NSCs with the lentivirus encoding green fluorescent protein (GFP) as described in the Materials and Methods section. Fluorescent labeling of NSCs with GFP (GFP-NSCs) was used to track transplants in the host brain. We showed that the established cell line GFP-NSCs stably expressed SOX2, a persistent marker for neural stem cell multipotency, as the native NSCs (Figure 1b). Once plated to adherent culture dishes and incubated with neural differentiation medium, both GFP-NSCs and the native NSCs differentiated into MAP2-positive neurons and GFAP-positive astrocytes (Figure 1c). We then examined the cytotoxicity of proinflammatory cytokines and Aβ on cultured NSCs using MTT assay. We found that cell viability of NSCs was markedly reduced by 48 h incubation with TNF-α, IL-6, IL-1β, Aβ_1–40_ or Aβ_1–42_, respectively (Figure 1d). 

### 2.2. Age-Associated Elevation of the AEP and Proinflammatory Cytokines in AD Mice

It has been reported that the AEP is upregulated in AD brains and could be a potential therapeutic target. We here chose the APPswe/PS1dE9 transgenic mice (APP/PS1) as an AD model, collected whole brain tissue and measured the enzymatic activity of AEPs during mice aging. Brain tissue analysis revealed that the AEP activity in 4-month-old APP/PS1 mice did not gain a significant increase in comparison with age-matched WT control mice, while from 5 to 8 months of age, the enzymatic activity of brain AEPs was significantly elevated in APP/PS1 mice compared to that in age-matched WT mice (Figure 2a). We analyzed the expression of the proinflammatory cytokines TNF-α, IL-6 and IL-1β in mouse brains with quantitative RT-PCR and ELISA assay (Appendix A). The RT-PCR analysis showed that 5-month-old APP/PS1 and WT mice exhibited similar transcript levels of TNF-α, IL-6 and IL-1β. At the age of 6 months, APP/PS1 mice began to show significantly enhanced transcription of TNF-α and IL-6 in the brain. From 7 to 8 months of age, the transcript levels of the three cytokines were all significantly upregulated in APP/PS1 compared to WT mice. Correspondingly, the ELISA revealed an aging-associated elevation in the protein levels of TNF-α, IL-6 and IL-1β in APP/PS1 mice brains (Figure 2b). These data indicated that there is an aging-associated upregulation of AEP activity and activation of neuroinflammatory response in the brain microenvironment of APP/PS1 mice. 

### 2.3. Inhibition of the AEP Reduced Brain Neuroinflammation and Aβ Production

Previous studies suggest that the AEP plays an important role in neuroinflammatory reaction. Therefore, we checked whether AEP activity was involved in this aging-associated activation of neuroinflammation in the AD brain. We employed a brain-penetrant AEP inhibitor named δ-secretase inhibitor 11 [39,44], which has an IC_50_ value of 0.31 ± 0.15 μM to suppress AEP activity in vitro (Figure 3a). We chose 7-month-old APP/PS1 mice and treated them with δ-secretase inhibitor 11 (10 mg kg^−1^, p.o., once daily) for one month and found that brain AEP activity was reduced (Figure 3b). The APP/PS1 mice is an AD model that overexpresses Aβ toxicity; AEPs were reported to mediate amyloidogenic processing of APP and Aβ over-generation. Hence, we checked the effect of AEP inhibition on Aβ production in brains of APP/PS1 mice. We found that 1 month oral administration of δ-secretase inhibitor 11 markedly decreased the secretion of Aβ_1–40_ and Aβ_1–42_ in the brain lysates (Figure 3c). Additionally, we found a significant decrease in the transcript and secretion levels of TNF-α, but not in IL-6 and IL-1β (Figure 3d,e). We extended this AEP inhibition treatment to 2 months, obtaining an effective reduction in the transcription and secretion of IL-6 and IL-1β (Figure 2f,g). These data suggest that the AEP plays a role in aging- and AD-associated inflammatory responses. Additionally, the tissue microenvironment in AD mice brain can be modified through inhibiting the AEP activity pharmacologically.

### 2.4. 1 Month Treatment with the AEP Inhibitor Did Not Improve Systemic Efficacy

We showed that 1 month of treatment with the AEP inhibitor only reduced TNF-α and Aβ generation in APP/PS1 mice. We next examined whether this 1-month treatment could achieve a systemic level of therapeutic efficacy by subjecting animals to the Morris water maze test. We found that there was no difference in swimming speed between age-mated WT mice and APP/PS1 mice which had received either vehicle or δ-secretase inhibitor 11 for one month (Figure 4a). Their escape latency decreased progressively over the 5-day training period, indicating a process of spatial learning and memory. Mice treated with either vehicle or δ-secretase inhibitor 11 all learned slower than WT mice (*p* < 0.05, Figure 4b). In the probe test, APP/PS1 mice treated with δ-secretase inhibitor 11 did not demonstrate better performance than vehicle-treated mice in the number of crossing platforms and time staying in target quadrant. Both of these two groups performed worse than the WT mice group (*p* < 0.05, Figure 4c,d). These data indicated that the 1-month drug treatment did not rescue cognitive impairment of APP/PS1 mice. We moved to the next step of the experiment: transplantation of NSCs into brains of APP/PS1 mice that had received 1 month of treatment with δ-secretase inhibitor.

### 2.5. Transplantation of NSCs in Combination with AEP Inhibition as a New Therapy

The 8-month-old APP/PS1 mice that had received 1 month of treatment with the AEP inhibitor (δ-secretase inhibitor 11) were divided into two groups: APP/PS1 + inhibitor and APP/PS1 + inhibitor + NSCs (Figure 4e). Similarly, vehicle-treated mice were divided into two groups: APP/PS1 + vehicle and APP/PS1 + NSCs. We transplanted GFP-NSCs into the bilateral hippocampus of APP/PS1 mice and continued to administer the AEP inhibitor for one more month. Two weeks after transplantation, a portion of mice were perfused and decapitated, and their brains were fixed. Histological examination of brain sections disclosed the survival of grafted GFP-NSCs in the hippocampus host brains (Figure 5a). We counted the number of GFP-NSCs in the hippocampus of each mouse and calculated their survival rates based on injected cell amount. In mice transplanted with NSCs alone, the survival rate of grafted NSCs was just over 0.5%, while pre-treatment with the AEP inhibitor nearly doubled the survival rate of NSCs to 1% in brains of APP/PS1 mice (Figure 5b), suggesting a supportive effect on stem cell survival resulting from pharmacological inhibition of the AEP.

### 2.6. Characterization of Cognitive Function and Hippocampal Neural Plasticity

One month after cell transplantation, the remaining mice were subjected to the Morris water maze test. The five groups of mice, including WT, APP/PS1 + vehicle, APP/PS1 + vehicle + NSCs, APP/PS1 + inhibitor and APP/PS1 + inhibitor + NSCs, showed no difference in swimming speed (Figure 6a). The escape latency is an indicator that reflects the process of spatial learning and memory. We found that mice treated either with NSCs alone or with AEP inhibitor alone all learned slower than WT mice (*p* < 0.05, Figure 6b), while mice receiving NSCs in combination with AEP inhibitor displayed a learning ability similar to WT. In the probe test, the number of crossing platforms showed that APP/PS1 mice receiving NSCs + AEP inhibitor performed better than mice treated with the AEP inhibitor alone (*p* < 0.05, Figure 6c). Additionally, AD mice receiving NSC transplantation + AEP inhibitor performed better than treatment with NSC transplantation alone (*p* < 0.05, Figure 6d). These data suggest that NSC transplantation combined with AEP inhibition gained improved therapeutic efficacy compared to either NSC transplantation alone or AEP inhibition alone. 

To evaluate the effect of NSC transplantation on hippocampal neural circuits, we conducted the electrophysiological characterization of synaptic plasticity in the hippocampus. The basal synaptic transmission was reflected by recording the field excitatory post-synaptic potentials (fEPSPs) in the CA1 region responding to stimulation of the Schaffer collaterals. The long-term potentiation (LTP) of fEPSPs, which represents the cellular and molecular basis of learning and memory, was induced by a theta-burst-stimulation described in the Methods section. We found that the magnitude of LTP in APP/PS1 mice receiving NSCs + AEP inhibitor was significantly higher than treatment with NSCs alone (Figure 6e). This suggests that NSC transplantation combined with AEP inhibition substantially prevented synaptic deterioration, which probably underlies the improved therapeutic efficacy of APP/PS1 mice. 

## 3. Discussion

In AD brains, deposits of Aβ and neuroinflammation are the major deleterious factors that continuously poison the brain microenvironment and damage engraftments [17,24]. Based on recent advances in AD pathology research, this study employed an AEP inhibition strategy to reduce the toxicity of the host brain microenvironment and incorporated it with transplantation of NSCs as a new therapy. Our data showed that proinflammatory cytokines and Aβ were toxic to the cell viability of NSCs. Pharmacological inhibition of the AEP activity in brains of APP/PS1 mice effectively suppressed age-associated inflammatory responses and Aβ production. The ameliorated tissue microenvironment improved the survival rate of implanted NSCs; thereby, this combination strategy gained better therapeutic efficacy than NSC transplantation alone or AEP inhibition alone. Brain slice recording indicated an improved functional plasticity in the hippocampus of combined therapy, which could be a cellular basis of improvement of the learning and memory function. Inhibition of the AEP alone did not gain systemic therapeutic effect in AD mice, although it antagonized Aβ and neuroinflammation. This outcome is likely related to the short period of treatment. Generally, experimental drug treatment for AD mice lasts 3 months, while the period in the current study was 1–2 months. 

The APP/PS1 transgenic mice burden two genetic mutations that lead to overexpression of APP and elevated Aβ levels [47]. In their brains, we found an age-associated increase in AEP activity and expression of proinflammatory cytokines (TNF-α, IL-6 and IL-1β). Elevation of AEP in AD and aged brains has been documented by many studies. AEP was reported as an upstream secretase that facilitates amyloidogenic APP processing and enhances Aβ secretion, i.e., an important player for disease onset of AD [36,37,38,39,40,41,42,43,44]. It is unsurprising that inhibition of AEP can decrease Aβ generation. The age-associated activation of neuroinflammation probably resulted from Aβ-triggered microglial inflammatory response. The oligomers of Aβ are strong stimulators that can induce microglial response and secretion of proinflammatory cytokines, likely through binding Toll-like receptors TLR2, TLR4 and TLR6 [48]. The AEP promotes amyloidogenic APP processing and Aβ production; in turn, inhibition of the AEP is certainly able to repress Aβ-induced microglial inflammatory response. 

In addition to Aβ oligomers, aging can also stimulate inflammatory response in microglia. Transcriptomic studies of microglia have identified upregulation of a host of age-related inflammatory genes in aged brains of human, APP/PS1 and other transgenic mice [49,50,51,52,53,54,55]. Currently, the molecular basis of age-activated microglial inflammatory reaction has not been revealed in detail [55]. Given that upregulation and activation of the AEP is closely related to brain aging, and the AEP was recently found to be involved in acute brain injury-induced neuroinflammation [45,46], it is of great interest to explore whether the AEP participates in an age-induced inflammatory response and microglia activation.

Although it is not completely clear how the AEP participates in inflammatory responses, researchers have suggested the advantage of targeting the AEP as an AD modification [36,37,39]. The expression level of AEPs in normal healthy brains is much lower than in spleen, colon, kidney, placenta, bladder, lung, thyroid and lymph node [36,39]. Upregulation of the brain AEP mostly occurs under neurodegenerative or acute damage conditions, which include aging, AD, PD, TBI, stroke and even glioblastoma [36]. Conceivably, pharmacological inhibition of brain AEPs does not necessarily compromise AEP function in peripheral organs, and is less likely to cause serious side effects. Our research continues to disclose the potentials of AEP inhibition in amelioration of the pathological microenvironment that causes cell death in AD brains, as well as in the secondary injury of TBI and stroke. 

Stem-cell-based therapy, with great potential in rehabilitation of neurodegenerative diseases such as AD and Parkinson’s disease (PD), has continuously been studied by scientists and has attracted the attention of doctors and patients [16,18]. Preclinical studies on NSC transplantation have pointed out that grafted NSCs could differentiate into neurons and/or glial cells and replace damaged cells in brains of neurodegenerative diseases [18]. After transplantation, NSCs may survive, differentiate and integrate into the host neural circuitry in the hippocampus, thereby restoring impaired synaptic plasticity and rescuing cognitive deficits. However, from the perspective of clinical application, the progress of stem cell therapy for brain disorders is still at an early stage. There are many challenges that hinder cell therapy becoming reality, such as, but not limited to, the selection and purification of stem cells, proliferation and neural differentiation efficiency [11,16,18]. After being transplanted, stem cells face another major obstacle: the cytotoxic microenvironment that is produced by detrimental factors in the host brain and leads to low graft survival of transplants [16,21].

This study provided a new idea to promote clinical translation of stem-cell-based therapy of Alzheimer’s disease (AD): transplantation of stem cells in combination with the pharmacological inhibition of the AEP; the latter is to antagonize Aβ toxicity and neuroinflammation in AD brains, where the AEP plays a critical role in disease progression. We demonstrated that improvement of the host brain microenvironment substantially increased graft survival in the brain of AD mice and improved efficacy of cell therapy. Stem cell transplantation combined with manipulation of brain microenvironment could be a promising therapeutic strategy for AD and other neurodegenerative diseases. 

## 4. Materials and Methods

### 4.1. Reagents and Antibodies 

The AEP inhibitor, also named δ-secretase inhibitor 11 (7-Morpholin-4-yl-benzo [1,2,5]oxadiazol-4-ylamine, PubChem CID: 1095027) [37,45], was purchased from J&K Scientific Ltd. (Beijing, China). The δ-secretase inhibitor 11 was first dissolved in dimethyl sulfoxide (DMSO) to make a stock solution and then diluted in a 0.9% NaCl solution containing gum arabic for systemic treatment. The ELISA kits for detecting IL-6 (70-EK206/3-96) and IL-1β (70-EK201B/3-96) were obtained from MULTISCIENCES Lianke Biotech, Co. Ltd. (Hangzhou, China). The ELISA kit for TNF-α (H052-96T) was obtained from Nanjing Jiancheng Bioengineering Institute (Nanjing, China). The primary antibody for mature AEP was obtained from R&D Systems Inc. (Minneapolis, MN, USA). Antibody for microtubule-associated protein 2 (MAP2) was obtained from Abcam (Cambridge, MA, USA). Antibodies for SOX2 (sc-365823) and Nestin (sc-23927) were obtained from Santa Cruz Biotechnology, Inc. (Santa Cruz, CA, USA). Isoflurane gas was purchased from RWD Life Science (Shenzhen, China).

### 4.2. Animals and Ethical Statements 

Animal studies are reported in compliance with the ARRIVE guidelines [56]. Male APPswe/PS1dE9 transgenic mice (APP/PS1) and their wild-type (WT) littermates were purchased from Changzhou Cavens Laboratory Animal Co., Ltd. (Changzhou, China). Male C57BL/6J mice were from Shanghai SLAC Laboratory Animal Co., Ltd. (Shanghai, China). Animals were housed in the pathogen-free animal facility at Shanghai Jiao Tong University School of Medicine (SJTU-SM). Mice were under a 12 h/12 h light/dark cycle and at 24 ± 2 °C, with free access to water and a standard rodent diet. Animal experimental procedures was approved by the Animal Experimentation Ethics Committee and Institutional Animal Care and Use Committee (IACUC) at SJTU-SM, and carried out strictly in accordance with the guideline of Association for Assessment and Accreditation of Laboratory Animal Care (AAALAC). To perform the AEP inhibition treatment, 7-month-old APP/PS1 mice were randomly divided into two groups: δ-secretase inhibitor 11 and vehicle treatment. The δ-secretase inhibitor 11 was given to mice once daily via oral gavage (10 mg kg^−1^) for 1 or 2 months. To collect brain tissue, mice were deeply anesthetized with 4% isoflurane (RWD Life Science, Shenzhen, China) and then decapitated.

### 4.3. Enzymatic Activity Assay 

Recombinant mouse AEP (R&D Systems, Inc., Minneapolis, MN, USA) was diluted to 50 μg mL^−1^ in activation buffer (0.1 M NaOAc, 0.1 M NaCl, pH 4.5) and incubated for 6 h at 37 °C with or without AEP inhibitor, then diluted to 2 ng μL^−1^ in assay buffer (50 mM MES, 250 mM NaCl, pH 5.5). Next, 50 μL of 2 ng μL^−1^ AEP was loaded in the plate, and the reaction was started by adding 50 μL of 200 μM Substrate: Z-Ala-Ala-Asn-AMC (Bachem AG, Bubendorf, Switzerland). Substrate was also diluted with assay buffer, including a Substrate Blank containing Assay Buffer and Substrate. This was read at excitation and emission wavelengths of 380 nm and 460 nm (top read), respectively, in kinetic mode for 45 min. Tissue homogenates (10 μg) were incubated in 200 μL assay buffer containing 20 μM AEP Substrate and assayed as above description. The activity of AEP was expressed as the reading at 45 min minus the first reading.

### 4.4. RNA Isolation and Quantitative Real-Time PCR

Brain tissue cells were lysed and processed for RNA extraction using a simply P total RNA extraction kit (BioFlux BSC21M1). cDNA was generated using a PrimeScript™ RT Master Mix (Takara RR036A). The ratio of reverse transcription system mixture was: 5XMix 2 μL, mRNA sample 3 μL and RNase Free H_2_O 5 μL. Other steps followed the manufacturer’s instructions. All the quantitative real-time PCR reactions were performed using a LightCycler 480II instrument (Roche, Basel, Switzerland). All samples were run in triplicate in a 96-well plate. The primers of cytokines are shown in Table 1. Dissociation curves were used to confirm amplification of a single product for each primer pair per sample. Expression levels for each gene were calculated as relative expression of target gene (2^(−ΔΔCT)^). GAPDH mRNA level was used as an internal reference. 

### 4.5. ELISA 

To measure Aβ concentration, the mouse brain tissue was homogenized in buffer (5 M guanidine HCl diluted in 50 mM Tris-HCl, pH 8.0) and incubated at room temperature for 3 h. Then, the samples were diluted with cold reaction buffer (phosphate-buffered saline with 5% BSA and 0.03% Tween 20, supplemented with protease inhibitor cocktail) and centrifuged at 16,000× *g* for 20 min at 4 °C. The supernatants were assayed by human Aβ_1–40_ and Aβ_1–42_ ELISA kits (#KHB3481 and #KHB3544, Invitrogen, Carlsbad, CA, USA) according to the manufacturer’s instructions. 

### 4.6. Isolation and Culture of Neural Stem Cells (NSCs) 

Pregnant mice were sacrificed on days 13–15 of pregnancy. Under aseptic conditions, the fetal mice brains were carefully obtained. Brain hemisphere was carefully isolated and dissociated with accutase (Stemcell Technologies Inc., Shanghai, China). After centrifugation and filtration purification, the isolated neural stem cells (NSCs) were then cultured in NeuroCult^TM^ Basal Medium (Mouse & Rat) supplemented with 10% NeuroCult^TM^ Proliferation Supplement, with 0.1% mouse recombinant EGF at 10 μg/mL, 0.2% human recombinant bFGF at 10 μg/mL and 0.1% Heparin solution at 37 °C in a 5% CO_2_ incubator. Medium was changed every two days. NSCs divided and proliferated in the form of neurospheres. 

### 4.7. Neural Differentiation of NSCs

We coated a 24-well glass bottom plate with the biolaminin substrates (10 μg/mL, BioLamina) before seeding the cells, collected the neurospheres and centrifuged; then, NSCs were pipetted into single cells and re-plated in 24-well glass bottom plates for differentiation. We used differentiation medium (NeuroCult^TM^ Basal Medium (Mouse & Rat) supplemented with 10% NeuroCult^TM^ Differentiation Supplement (Stemcell Technologies Inc.) for continuous culture over 8 days to induce differentiation of NSCs. During this period, the medium was changed in time according to the cell status.

### 4.8. Transfection of the Green Fluorescent Protein (GFP) into NSCs

The plasmid Ubi-MCS-3FLAG-CBh-gcGFP-IRES-puromycin was packaged into lentivirus (Genechem Technologies Inc., Shanghai, China) to introduce the green fluorescent protein (GFP) into NSCs. Neurospheres were dissociated into single cells with accutase (Stemcell Technologies Inc., Vancouver, BC, Canada.) and seeded to 6 cm diameter culture dishes at a concentration of 1 × 10^6^ cell/dish. Dishes were coated with biolaminin (10 μg/mL, BioLamina, Sundbyberg, Sweden). After 24 h, NSCs were infected with the lentivirus according to the manufacturer’s instructions. Two days after lentiviral transfection, GFP-positive NSCs were screened in medium containing puromycin (1 μg/mL) and sorted with a FACS Arial cell sorter (BD Biosciences, San Jose, CA, USA). GFP-NSCs were then propagated for at least 5 passages in NSC medium and transplanted into mice hippocampus.

### 4.9. Transplantation of GFP-NSCs

Neurospheres of GFP-NSCs at passage 5 were dissociated into single cells with accutase (Stemcell Technologies Inc.) and re-suspended with DPBS to a cell concentration of 5 × 10^4^/μL for transplantation. Concurrently, APP/PS1 mice were anesthetized with inhalation of isoflurane gas (RWD Life Science) which was balanced with oxygen and dialed to 2.0% for the induction of anesthesia and 1.0% for maintenance. Then, mice were mounted on a stereotaxic apparatus (RWD life science). Mouse head skin was shaved and sterilized with an iodine complex. The scalp was dissected along the midline and retracted bilaterally to expose the sagittal and coronal sutures. Bilateral hippocampal DG regions were selected as the implant sites following coordinates relative to Bregma: AP, −2.18 mm; ML, ±1.50 mm; DV, −2.30 mm. We used a 10 μL Hamilton micro-syringe to inject 4 μL cell suspension containing 2 × 10^5^ GFP-NPCs at a rate of 0.4 μL/min into each side of the hippocampus. After transplantation, the scalp of head was sutured and mice were kept in a warm incubator until waking.

### 4.10. Morris Water Maze Test 

Mice were trained in a round water pool (120 cm diameter) with extra-maze cues. Each animal received 4 training trials per day for 5 consecutive days to learn to find the hidden platform located 1.5 cm below the water surface. In each trial, mice were given 60 s to find the invisible platform in one of four different positions. The escape latency (the time required to find and climb onto the platform) recorded by Noldus Ethovision XT software version 11.5 (Beijing, China) was up to 60 s. After each trial, mice were dried and kept in a warm cage. The probe test was conducted 24 h after the last training. Water maze test data were analyzed by an investigator who was blinded to the treatment.

### 4.11. Hippocampal Slice Preparation and Electrophysiological Recording

APP/PS1 mice were anaesthetized with isoflurane, decapitated, and their brains were dropped in ice-cold artificial cerebrospinal fluid (a-CSF) containing 124 Mm NaCl, 3 mM KCl, 1.25 mM NaH_2_PO_4_, 6.0 mM MgCl_2_, 26 mM NaHCO_3_, 2.0 mM CaCl_2_ and 10 mM glucose. The hippocampi were dissected and cut into 400 μm thick transverse slices with a vibratome (VT 1000S, Leica Inc., Wetzlar, Germany). After incubation at room temperature (23–24 °C) in a-CSF for 60–90 min, slices were placed in a recording chamber (RC-22C, Warner Instruments) on a fixed stage upright microscope for electrophysiology (BX51 WI with water immersion objective lens, Olympus, Japan). Slices were perfused at a rate of 3 mL/min with a-CSF at 23–24 °C. The Schaffer collaterals were stimulated with a tungsten monopolar electrode (0.1 MΩ) and the field excitatory post-synaptic potentials (fEPSPs) were recorded in CA1 stratum radiatum by a glass microelectrode filled with a-CSF with resistance of 3–4 MΩ. The stimulation output was controlled by an electric stimulator (Master-8; AMPI, Jerusalem). fEPSPs were recorded under current-clamp mode of a Multiclamp 700B amplifier (Molecular Devices, San Jose, CA, USA). The stimulus intensity (0.1 ms, 3.0–4.5 V) was set to evoke 40% of the maximum f-EPSP and the test pulse was applied at a rate of 0.033 Hz. A theta-burst stimulation (TBS, 4 pulses at 100 Hz, repeated 3 times with a 200 ms interval) was used to induce LTP of fEPSPs. The magnitudes of LTP were the mean percentage of baseline fEPSP initial slope.

### 4.12. Histological Examination of Brain Sections 

Mice were deeply anesthetized and perfused with 4% paraformaldehyde (PFA) in phosphate-buffered saline (PBS) through the heart. The brains were removed and cut into coronal sections with 20 μm thickness using a cryostat microtome (Leica CM1950, Germany) and stored at –80 °C until staining. Permeabilization of brain sections was performed in PBS with 0.1% Triton X-100 for 15 min at room temperature. After being blocked, sections were incubated with primary antibody overnight at 4 °C. After washing three times with PBS, sections were incubated with Alexa Fluor 488-conjugated donkey anti-rabbit or Alexa Fluor 594 anti-mouse IgG secondary antibodies (Invitrogen, Carlsbad, CA, USA). Photographs were taken and analyzed using a Leica SP8 confocal microscope (Leica Microsystems, Wetzlar, Germany). Quantification of grafted GFP-NSCs was carried out in six slices of each brain spaced 120 μm apart to estimate the average cell number per unit area. Quantification and analysis was conducted by a person who was blinded to the treatment.

### 4.13. Statistical Analysis 

Data were expressed as mean ± SEM and analyzed using Prism 7 software (La Jolla, CA, USA). The concentration of the inhibitor yielding half-maximal inhibition (IC_50_) of AEP activity was calculated using the equation: Fractional Enzymatic Activity (% of control) = Bottom + (Top-Bottom)/(1 + 10^((LogIC_50_-C) *n)), where C is the logarithm of inhibitor concentration and n is the Hill coefficient. The statistical difference between two independent groups was analyzed using unpaired Student’s *t*-test. Additionally, the differences among more than two groups was assessed by the parametric one-way ANOVA followed by a Tukey’s post hoc test. For the Morris water maze test, a two-way ANOVA and repeated measures was used to compare acquisition data of different groups. Differences were considered to be significant when *p* < 0.05.

## Figures and Tables

**Figure 1 ijms-24-07739-f001:**
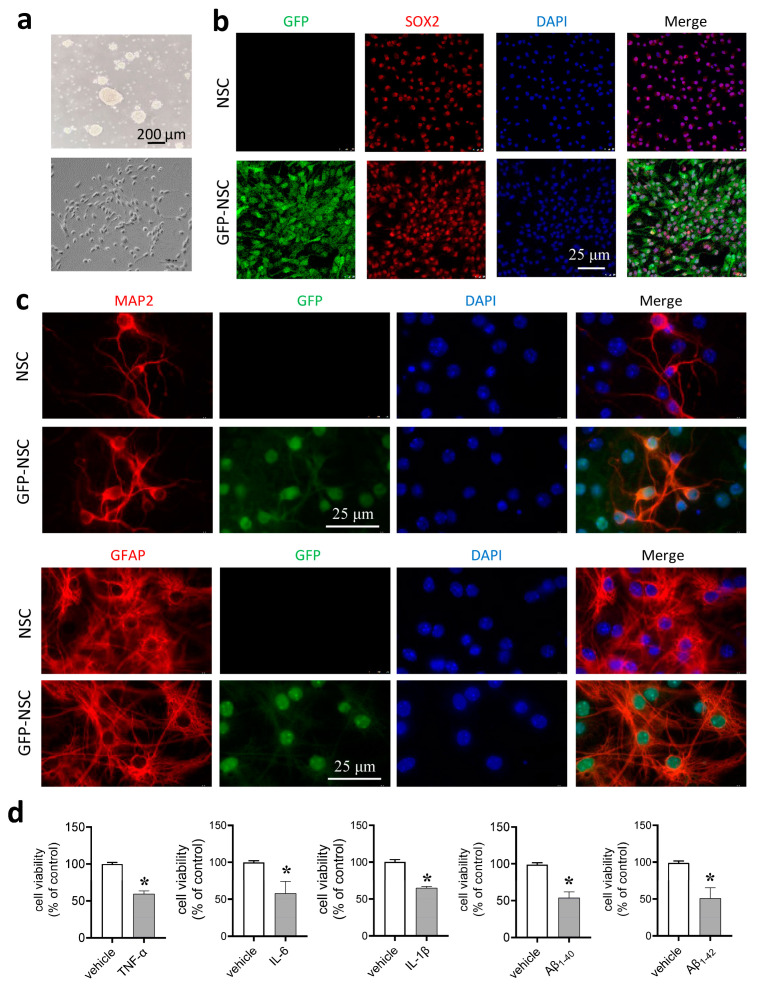
Mouse neural stem cells (NSCs) transfected with green fluorescent protein (GFP). (**a**) Bright field images of neurospheres in the suspension culture and dispersed NSCs in the adherent monolayer culture. Scale bar, 200 μm. (**b**) NSCs and GFP-NSCs stably expressed SOX2, a persistent marker for neural stem cell multipotency. Scale bar, 25 μm. (**c**) GFP-NSCs and the native NSCs differentiated into neurons (MAP2 positive) and GFAP positive astrocytes. Scale bar, 25 μm. (**d**) Cell viability of NSCs after a 48 h incubation with TNF-α (10 ng/mL), IL-6 (10 ng/mL), IL-1β (50 ng/mL), Aβ_1–40_ (15 μM) and Aβ_1–42_ (15 μM), respectively. * *p* < 0.05, unpaired Student’s *t*-test; *n* = three independent experiments.

**Figure 2 ijms-24-07739-f002:**
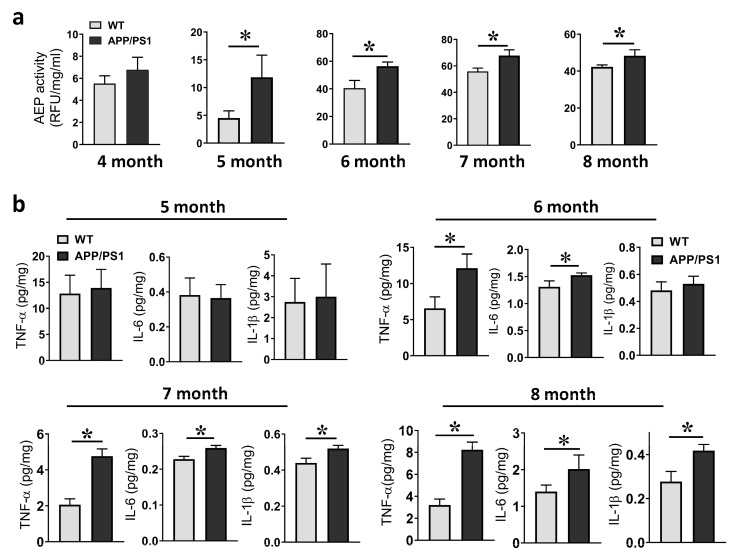
The enzymatic activity of the asparaginyl endopeptidase (AEP) and proinflammatory cytokines in whole brain tissue of mice. (**a**) Activity of AEPs detected in brains of WT and APP/PS1 mice, 4 to 8 months old. * *p* < 0.05, unpaired Student’s *t*-test; *n* = 4–5 mice per group. (**b**) Expression of TNF-α, IL-6 and IL-1β in brains of 5-to-8-month-old mice. * *p* < 0.05, unpaired Student’s *t*-test; *n* = 4–5 mice per group.

**Figure 3 ijms-24-07739-f003:**
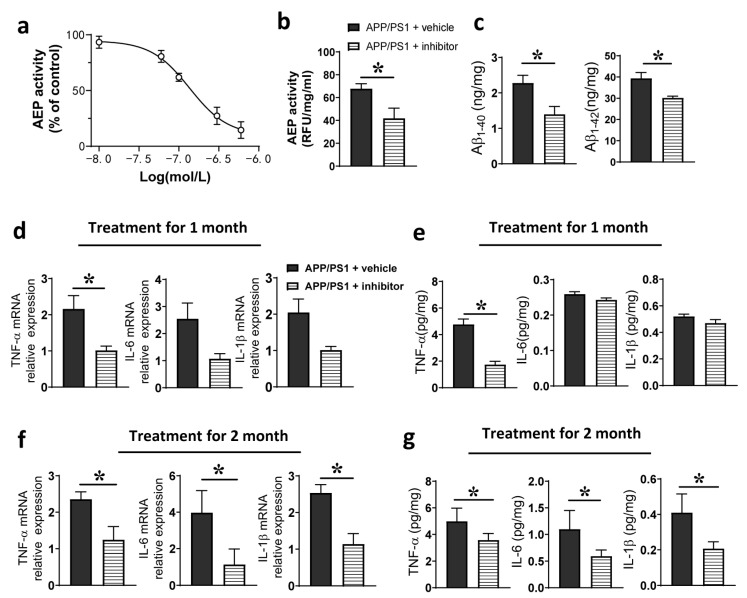
Pharmacological inhibition of the AEP with δ-secretase inhibitor 11. (**a**) The concentration-response of δ-secretase inhibitor 11 on AEP activity, *n* = 5 tests. (**b**) Brain AEP activity of APP/PS1 mice after one month of treatment with vehicle or δ-secretase inhibitor 11. * *p* < 0.05, unpaired Student’s *t*-test; *n* = 5 mice per group. (**c**) Aβ_1–40_ and Aβ_1–42_ in brains of APP/PS1 mice treated with vehicle or δ-secretase inhibitor 11 for one month. * *p* < 0.05, unpaired Student’s *t*-test; *n* = 4–5 mice per group. (**d**,**e**) Transcript and secretion levels of TNF-α, IL-6 and IL-1β in brains of APP/PS1 mice after one-month treatment with δ-secretase inhibitor 11. * *p* < 0.05. (**f**,**g**) TNF-α, IL-6 and IL-1β levels after two-month treatment with δ-secretase inhibitor 11. * *p* < 0.05, unpaired Student’s *t*-test; *n* = 4–5 mice per group.

**Figure 4 ijms-24-07739-f004:**
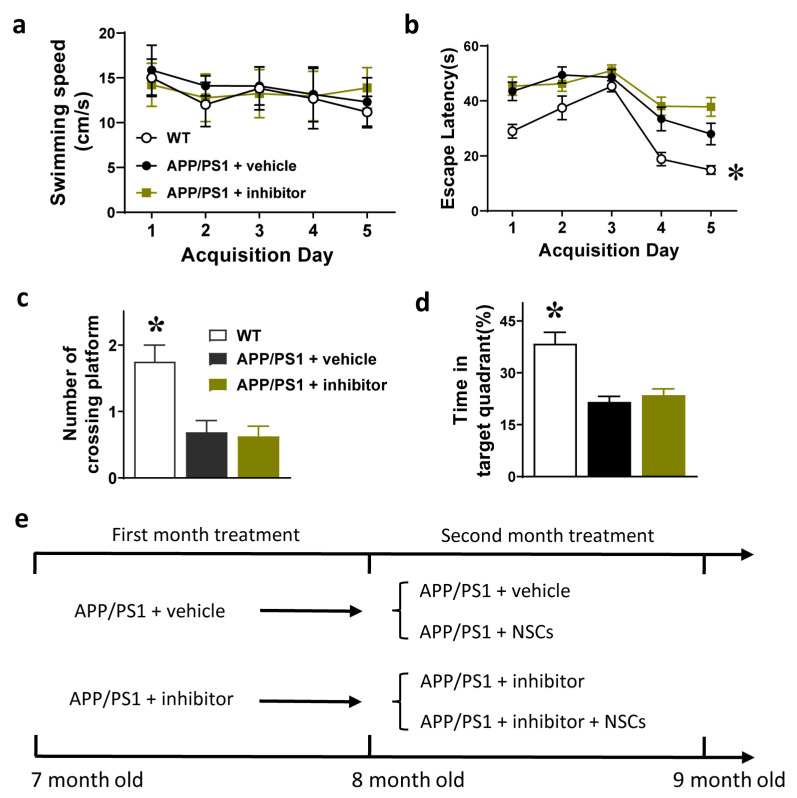
One month of AEP inhibition treatment and the cognitive function of APP/PS1 mice. (**a**) Over the 5-day acquisition training, the 3 groups of mice showed similar swimming speed. (**b**) The escape latency of WT mice compared with APP/PS1 mice treated with either δ-secretase inhibitor 11 or vehicle (*p* < 0.05, two-way ANOVA analysis, *n* = 16 mice per group). (**c**,**d**) The number of crossing platforms and time spent in the target quadrant (probe test), which was performed on day 6 after the acquisition training. * *p* < 0.05, WT mice compared with APP/PS1 mice treated with either δ-secretase inhibitor 11 or vehicle, one-way ANOVA followed by a Tukey’s post hoc test, *n* = 16 mice per group. (**e**) For cell transplantation experiment, the 8-month-old APP/PS1 mice that had received 1 month of treatment with vehicle or the AEP inhibitor were divided into 4 groups: (1) APP/PS1 + vehicle, (2) APP/PS1 + NSCs, (3) APP/PS1 + inhibitor and (4) APP/PS1 + inhibitor + NSCs. The treatment was planned to be extended for one more month. *n* = eight mice per group.

**Figure 5 ijms-24-07739-f005:**
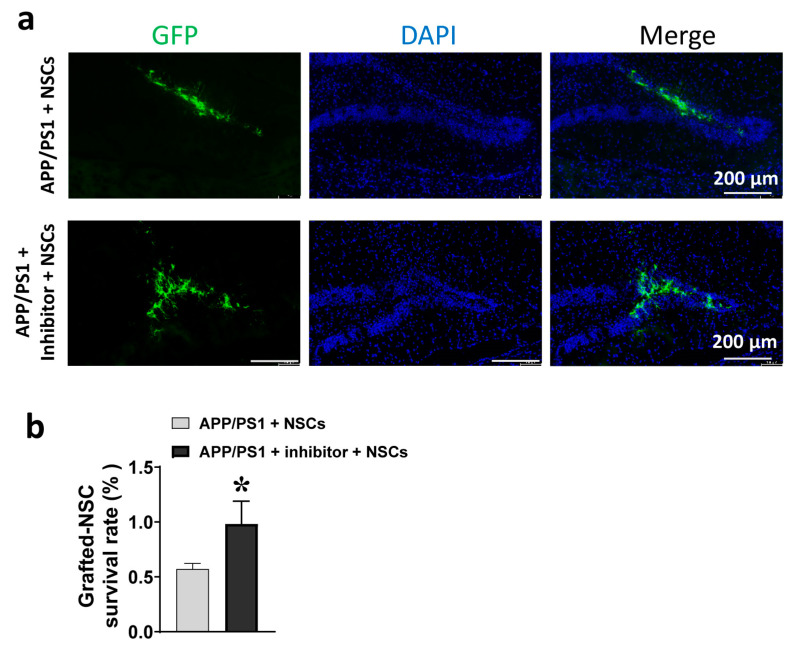
Survival of NSCs two weeks after implantation into the hippocampus of APP/PS1 mice. (**a**) GFP expressing NSCs (GFP-NSCs) were tracked in the injection sites of AD mice with or without pre-treatment with the AEP inhibitor. Scale bar, 200 μm. (**b**) Survival rate of NSCs engrafted in brains of the two group mice. * *p* < 0.05, unpaired Student’s *t*-test; *n* = five mice per group.

**Figure 6 ijms-24-07739-f006:**
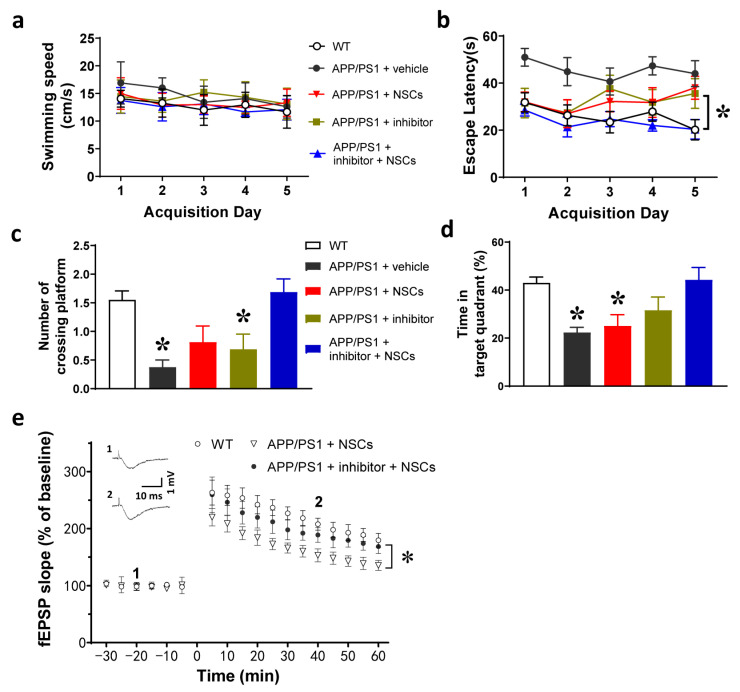
The cognitive function of APP/PS1 mice and neural plasticity in the hippocampus one month after cell transplantation. (**a**) Over the 5-day acquisition training, the 5 groups of mice showed similar swimming speed. (**b**) The escape latency of the five groups. * *p* < 0.05, APP/PS1 mice receiving NSCs + AEP inhibitor treatment compared with mice treated with either NSC alone or δ-secretase inhibitor 11 alone, two-way ANOVA analysis, *n* = eight mice per group. (**c**) The number of crossing platforms (probe test). * *p* < 0.05, APP/PS1 mice receiving NSCs + AEP inhibitor compared with mice treated with either vehicle or AEP inhibitor. (**d**) Time spent in the target quadrant (probe test). * *p* < 0.05, APP/PS1 mice receiving NSCs + AEP inhibitor compared with mice treated with either vehicle or NSCs alone. Probe test data were analyzed by one-way ANOVA and a Tukey’s post hoc test, *n* = eight mice per group. (**e**) Long-term potentiation (LTP) of fEPSPs induced in the hippocampus of AD mice. * *p* < 0.05, the LTP magnitude (from 5 to 60 min) was higher in AD mice receiving NSCs + AEP inhibitor than mice + NSCs alone (two-way ANOVA, *n* = 6 mice per group). Trace 1 and trace 2 are representative fEPSPs of WT mice recorded at the time point 1 and 2.

**Table 1 ijms-24-07739-t001:** Primer gene sequences.

Cytokines	Forward Primer (Mouse)	Reverse Primer (Mouse)
TNF-α	GAGTGACAAGCCTGTAGCCC	TTGTCCCTTGAAGAGAACCTG
IL-6	CTTGGGACTGATGCTGGTGA	ACTCTTTTCTCATTTCCACGATTT
I-1β	AAATCTCGCAGCAGCACAT	ATGAGTCACAGAGGATGGGC
GAPDH	GAGTGTTTCCTCGTCCCGTAG	AGGTCAATGAAGGGGTCGTT

## Data Availability

The data supporting the findings of this study are available within the article and from the corresponding author upon request.

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
