# Peer review of "Pharmacological Inhibition of the Asparaginyl Endopeptidase (AEP) in an Alzheimer’s Disease Model Improves the Survival and Efficacy of Transplanted Neural Stem Cells"

_ijms, 2023, doi:10.3390/ijms24097739_

Round 1

Reviewer 1 Report

This is an excellent study providing a novel approach to facilitating the survival and efficacy of transplanted neural stem cells (NSC) in a model of Alzheimer's disease (AD) in mice. The beauty of the study is to combine the inhibition of asparaginyl endopeptidase (AEP) for reducing systemic inflammation and amyloid-beta generation, which in isolation appears to be insufficient to have meaningful clinical effects, with cultured NCS to gain significant improvement in neurophysiological and functional outcome. 

The methodology is well-described and executed, the data is clear and the conclusions supported by the reported findings. Except for a few typographical errors, the manuscript is clearly written in excellent English and is very easy to follow. Although further studies are needed to corroborate the findings and also establish the efficacy of this approach in humans, this study does introduce a method that can have meaningful clinical effects in patients with AD diagnosis. I would like to congratulate the authors for their accomplishments and would be delighted to see this study published in IJMS. 

Minor comments:

LM 40-41: The statement refers to studies until 2017 but the cited reference was published in 2014

LM 60: Please define A-beta (only defined in the abstract)

LM 64: NSCs were greatly suppressed after [they were] transplanted into brains

LM 80: AEP is an effective modulation target that can reduce A-beta generation... This statement implies that AEP reduces A-beta generation.  Please rephrase (can for instance state that it can serve as a target for reducing A-beta generation)

LM 86: was found able to reduce ... (delete able)

LM 87: reporting the AEP as an important ... (delete the)

LM 94: we supposed (change to hypothesized)

LM 100: Hopefully, ... (change to We examined if ...)

LM 137: exhibited the similar ... (delete the)

LM 139: Delete While

Discussion: Consider changing the first paragraph to summarizing the main findings of the study (this information appears only at the end of the paragraph and after repeating some introductory statements). 

throughout the manuscript: Consider changing the Ageing (British) to Aging

Author Response

Responses to: Comments and Suggestions for Authors (1)

We are very grateful to the reviewer for the following comments and suggestions.

This is an excellent study providing a novel approach to facilitating the survival and efficacy of transplanted neural stem cells (NSC) in a model of Alzheimer's disease (AD) in mice. The beauty of the study is to combine the inhibition of asparaginyl endopeptidase (AEP) for reducing systemic inflammation and amyloid-beta generation, which in isolation appears to be insufficient to have meaningful clinical effects, with cultured NCS to gain significant improvement in neurophysiological and functional outcome.

The methodology is well-described and executed, the data is clear and the conclusions supported by the reported findings. Except for a few typographical errors, the manuscript is clearly written in excellent English and is very easy to follow. Although further studies are needed to corroborate the findings and also establish the efficacy of this approach in humans, this study does introduce a method that can have meaningful clinical effects in patients with AD diagnosis. I would like to congratulate the authors for their accomplishments and would be delighted to see this study published in IJMS.

Minor comments:

  • LM 40-41: The statement refers to studies until 2017 but the cited reference was published in 2014

Response: 2017 has been changed to 2014.

  • LM 60: Please define A-beta (only defined in the abstract)

Response: “Aβ oligomers” was changed to “the amyloid-beta (Aβ) oligomers”

  • LM 64: NSCs were greatly suppressed after [they were] transplanted into brains

  Response: Right, we added “they were” to this sentence.

  • LM 80: AEP is an effective modulation target that can reduce A-beta generation... This statement implies that AEP reduces A-beta generation. Please rephrase (can for instance state that it can serve as a target for reducing A-beta generation)

  We appreciate this suggest and modified the sentence as:

Recent accumulating evidence indicates that the lysosomal asparaginyl endopeptidase (AEP) or legumain can serve as an intervention target for reducing Aβ generation and inhibiting inflammatory responses [36-38].

  • LM 86: was found able to reduce ... (delete able)

   Right, we deleted “able”.

  • LM 87: reporting the AEP as an important ... (delete the)

   Yes, we deleted “the”.

  • LM 94: we supposed (change to hypothesized)

  Right, we changed “supposed” to “hypothesized”.

  • LM 100: Hopefully, ... (change to We examined if ...)

  Thank you so much. We changed it to “examined if…”.

  • LM 137: exhibited the similar ... (delete the)

   Yes, we deleted “the”.

  • LM 139: Delete While

   Right, we deleted “While”.

  • Discussion: Consider changing the first paragraph to summarizing the main findings of the study (this information appears only at the end of the paragraph and after repeating some introductory statements).

  This is a very good advice. We re-edited “the first paragraph” as below:

  1. Discussion

In AD brains, deposits of Aβ and neuroinflammation are the major deleterious factors that continuously poison the brain microenvironment and damage engraftments [17, 24]. Based on recent advances in AD pathology research, this study employed an AEP inhibition strategy to reduce toxicity of the host brain microenvironment and incorporated it with transplantation of NSCs as a new therapy. Our data showed that proinflammatory cytokines and Aβ were toxic to cell viability of NSCs. Pharmacological inhibition of the AEP activity in brains of APP/PS1 mice effectively suppressed age-associated inflammatory responses and Aβ production. The ameliorated tissue microenvironment improved survival rate of implanted NSCs, thereby this combination strategy gained better therapeutic efficacy than NSC transplantation alone or AEP inhibition alone. Bain slice recording indicated an improved functional plasticity in the hippocampus of combined therapy, which could be a cellular basis of improvement of the learning and memory function. Inhibition of the AEP alone didn’t gain systemic therapeutic effect in AD mice although it antagonized Aβ and neuroinflammation. This outcome is likely related to a short period of treatment. Generally, experimental drug treatment for AD mice lasts 3 months, while this period in current study was 1-2 months.

 [Please see the revised manuscript for the other part of discussion.]

  • Throughout the manuscript: Consider changing the Ageing (British) to Aging

Right. We replaced “Ageing” with “Aging” throughout this manuscript.

Reviewer 2 Report

The authors present a work related to the efficiency of reversion in Alzheimer's disease by injection of neuronal stem cells in concert with the effects induced by treatment with asparaginyl endopeptidase inhibitor and its role in the disease-related inflammation processes.

In general, the work is well presented and described, only a few specifics should be reported:

1) In the first sentence of the introduction the authors report ..... prodromal stage and dementia ONLY in late stage. It is suggested to replace only with SPECIALLY, this because even if in a low percentage this condition has been reported in early stages of AD.

2) Also in the introduction, it is somewhat surprising that no reference is made to the cytokine CX3CL1/Frakline, which has recently been reported to be directly involved in AD and to be represented in its soluble form in the cerebrospinal fluid of Alzheimer's subjects.

3) In the results, in particular in Figure 2-a, the enzymatic activity of AEP over time in WT and APP/PS mice is reported, together with the expression of TNF-alpha, IL-& and IL-1beta. From this analysis the authors decided to take 7 months as the reference time for the NSCs injection experiments; probably in consideration of the high expression of TNF-alpha but not of the other two interleukins analyzed with respect to WT. This choice should be more substantiated than an early or later stage (where an effect on all three inflammatory molecules is reported), as well as a minor difference in enzyme activity of AEP in Wt compared to APP/PS1.

4) Just to know, in paragraph 2.6 there is a typing error related to : cognitive function.

Author Response

Responses to: Comments and Suggestions for Authors (2)

We are very grateful to the reviewer for the following comments and suggestions.

The authors present a work related to the efficiency of reversion in Alzheimer's disease by injection of neuronal stem cells in concert with the effects induced by treatment with asparaginyl endopeptidase inhibitor and its role in the disease-related inflammation processes.

In general, the work is well presented and described, only a few specifics should be reported:

1) In the first sentence of the introduction the authors report ..... prodromal stage and dementia ONLY in late stage. It is suggested to replace only with SPECIALLY, this because even if in a low percentage this condition has been reported in early stages of AD.

Response:

Thank you. We followed this advice and replaced ONLY with specially.

2) Also in the introduction, it is somewhat surprising that no reference is made to the cytokine CX3CL1/Frakline, which has recently been reported to be directly involved in AD and to be represented in its soluble form in the cerebrospinal fluid of Alzheimer's subjects.

Response:

We added information regarding CX3CL1/Frakline to the introduction as:

Fractalkine, also named as C-X3-C Motif Chemokine Ligand 1 (CX3CL1), is produced by neurons while its receptor CX3CL1R is mainly expressed on microglia. The CX3CL1-CX3CR1 signaling was initially found to exhibit an anti-inflammatory effect by some AD pathology studies, but in some cases it instead incurred neurodegeneration [34-35].

3) In the results, in particular in Figure 2-a, the enzymatic activity of AEP over time in WT and APP/PS mice is reported, together with the expression of TNF-alpha, IL-& and IL-1beta. From this analysis the authors decided to take 7 months as the reference time for the NSCs injection experiments; probably in consideration of the high expression of TNF-alpha but not of the other two interleukins analyzed with respect to WT. This choice should be more substantiated than an early or later stage (where an effect on all three inflammatory molecules is reported), as well as a minor difference in enzyme activity of AEP in Wt compared to APP/PS1.

Response:

Thanks for your comprehension. In brains of APP/PS mice, elevation of the AEP activity and increase of inflammatory levels (TNF-α, IL-6, IL-1β) are all associated with aging. In the brain of 6-month APP/PS mice, the increase of IL-1β level didn’t yet reach a significant level. In many studies of Alzheimer’s disease (AD) employed APP/PS mice, 6-7 month old mice were often used for drug treatment. For example:

Tabassum Majid, et al. Pharmocologic treatment with histone deacetylase 6 inhibitor (ACY-738) recovers Alzheimer's disease phenotype in amyloid precursor protein/presenilin 1 (APP/PS1) mice. Alzheimers Dement (N Y). 2015 Oct 11;1(3):170-181. doi: 10.1016/j.trci.2015.08.001. eCollection 2015 Nov.

4) Just to know, in paragraph 2.6 there is a typing error related to : cognitive function.

Yes. We have corrected typing error in “function”.
